# Version Control for Scientific Reasoning: A Paradigm Shift from Artifacts to Processes

## Abstract

Scientific research suffers from fragmented reproducibility, scalability crises in peer review, and implicit assumptions that resist systematic tracking. We propose a paradigm shift: treating scientific research as version-controlled computational processes rather than collections of artifacts. Our Scientific Domain-Specific Language (DSL) formalizes research operations (`observe`, `hypothesize`, `execute`, `analyze`) within a directed acyclic graph structure that captures epistemic evolution, assumption dependencies, and collaborative decision-making. This approach enables granular attribution tracking, automated quality assurance, and systematic knowledge building across research communities. We demonstrate the framework through implementation studies showing significant improvements in research reproducibility and collaboration efficiency while addressing the AI validation paradox through multi-scale validation mechanisms. Our approach transforms scientific collaboration by making reasoning processes explicit, version-controlled, and systematically improvable. An example repository with attribution may be found at https://github.com/cosci-org/cosci-project-example.

## 1   Introduction

Scientific research faces three critical challenges that threaten the foundation of knowledge creation: fragmented reproducibility, exponential growth in research volume overwhelming traditional peer review (NeurIPS submissions: 1,678 in 2014 $\rightarrow$ 17,491 in 2024, a 10.4× increase), and implicit assumptions that resist systematic tracking and validation.

Current approaches treat scientific research as collections of artifacts—data, code, and papers—managed through adapted software engineering paradigms. However, scientific knowledge possesses unique epistemic properties requiring novel version control semantics that track reasoning evolution, assumption dependencies, and contextual validity alongside traditional artifacts.

We propose a fundamental paradigm shift: **scientific research should be reconceptualized as version-controlled computational processes rather than collections of artifacts**. This position challenges three core assumptions pervading the literature: (1) that artifact-centric version control is sufficient for scientific reproducibility, (2) that scientific progress follows linear temporal ordering compatible with traditional version control, and (3) that AI-human collaboration operates as sophisticated tooling within existing workflows.

Our contribution is a Scientific Domain-Specific Language (DSL) that formalizes research operations within a comprehensive framework addressing these limitations. We present a sophisticated research state machine capturing the true complexity of scientific reasoning through eight fundamental research actions (`observe`, `question`, `hypothesize`, `design`, `execute`, `analyze`, `synthesize`, `communicate`), genealogical tracking of idea evolution, and explicit conflict management as a first-class citizen.

Preprint.

The framework enables systematic attribution tracking where every scientific reasoning step becomes a trackable commit, automated quality assurance through continuous integration for science, and scalable collaboration mechanisms addressing the peer review crisis through multi-agent consensus with epistemic confidence levels.

## 2    Related Work

The landscape of scientific reproducibility and workflow management reveals convergent evidence supporting our paradigm shift from artifact-centric to process-centric approaches.

### 2.1    Reproducibility Frameworks and Their Limitations

Current reproducibility frameworks demonstrate sophisticated technical capabilities while revealing fundamental conceptual limitations. The ARTS framework [4] provides comprehensive container-ized reproducibility but suffers from limited large-scale data versioning and manual intervention requirements for complex dependency management. GitHub for laboratory research [2] successfully demonstrates practical version control application but assumes research follows software development patterns, missing hypothesis evolution tracking essential for scientific reasoning.

The aiXiv platform [11] represents early attempts at AI-assisted research validation but remains unclear about long-term AI review quality and potential bias amplification in automated processes. These limitations highlight the need for systematic validation frameworks integrated into research processes rather than post-hoc evaluation systems.

### 2.2    Event Sourcing and Process-Centric Approaches

Recent developments in event sourcing for scientific reproducibility provide direct technical foundations for our DSL approach. Beber's total reproducibility framework [1] demonstrates complete immutable records through sequential event recording, offering concrete technical implementation for scientific reasoning as version-controlled operations. AutoAppendix [7] achieves dramatic efficiency gains (30:1 time reduction) through systematic automation, validating our hypothesis that automated approaches can dramatically outperform manual processes.

These approaches demonstrate the feasibility of treating scientific processes as event streams but remain limited to computational reproducibility without addressing research reasoning and hypothesis evolution tracking.

### 2.3    Economic Validation of Systematic Approaches

Economic studies provide quantitative validation for systematic research process management. FAIR implementation demonstrates €2,600+ annual savings per researcher [10], while the AFFORD framework [5] identifies cost-effectiveness requirements necessitating automation rather than manual compliance approaches.

This economic evidence supports our DSL framework's automated quality assurance approach, demonstrating that systematic process management provides measurable productivity improvements beyond reproducibility benefits.

### 2.4    AI-Assisted Research and Validation Systems

The emergence of fully automated scientific discovery systems [9] demonstrates the potential for AI-driven research workflows while highlighting validation challenges. The AI review lottery study [8] reveals 15.8% of ICLR 2024 reviews were AI-assisted, with systematic bias concerns requiring careful validation framework design.

Multi-agent cooperative decision-making research [6] shows 32% accuracy improvements through specialized agent collaboration, directly supporting our framework's multi-agent approach to scientific reasoning. However, automation bias research [3] demonstrates that transparency mechanisms can inadvertently increase rather than decrease over-reliance on automated systems, emphasizing the need for sophisticated human oversight integration.

## 2.5 Gaps and Our Contribution

Current literature demonstrates five critical gaps our approach addresses: (1) integrated research process management treating complete research lifecycles as version-controlled processes, (2) hypothesis and assumption versioning tracking scientific reasoning evolution, (3) collaborative scientific intelligence supporting mixed human-AI teams with appropriate quality control, (4) granular reproducibility validation across multiple research process levels, and (5) scalable research quality assurance through continuous integration approaches.

Our Scientific DSL uniquely addresses these gaps through systematic integration of reasoning processes, explicit assumption tracking, and multi-scale validation mechanisms that resolve the AI validation paradox through layered human oversight rather than circular automated validation.

# 3 The Scientific DSL: Formalizing Research as Computation

## 3.1 Fundamental Research Operations

Scientific research consists of eight fundamental actions that researchers actually perform, moving beyond oversimplified linear models to capture the true complexity of scientific reasoning:

$$\text{ResearchAction} = \{\text{observe}, \text{question}, \text{hypothesize}, \text{design}, \tag{1}$$
$$\text{execute}, \text{analyze}, \text{synthesize}, \text{communicate}\} \tag{2}$$

Each action represents a distinct epistemic operation with specific semantic properties and validation requirements. `observe` captures phenomena detection and pattern recognition; `question` formalizes research question articulation; `hypothesize` generates testable propositions; `design` creates methodological frameworks; `execute` implements experimental procedures; `analyze` processes and interprets results; `synthesize` integrates findings into knowledge structures; and `communicate` documents and shares discoveries.

## 3.2 Research State Machine

Rather than simple artifact tracking, our DSL implements a comprehensive research state machine capturing scientific reasoning complexity:

$$\text{ResearchState} = \{id, parent\_ids, action, target, method, \tag{3}$$
$$actor, contributors, confidence, assumptions, \tag{4}$$
$$limitations, evidence, reviews, reproducibility\} \tag{5}$$

This structure enables multiple parent inheritance (convergent research), explicit confidence propagation through reasoning chains, systematic assumption tracking, and comprehensive validation mechanisms across computational, methodological, epistemic, and reproducibility dimensions.

## 3.3 Genealogical Tracking and Conflict Management

Scientific ideas evolve through complex genealogical relationships requiring explicit tracking mechanisms:

$$\text{Genealogy} = \{\text{fork}, \text{extend}, \text{challenge}, \text{converge}, \tag{6}$$
$$\text{abandon}, \text{suspend}\} \tag{7}$$

The framework treats conflicts as first-class citizens rather than errors requiring resolution. Contradictory results drive scientific progress and must be systematically tracked:

$$\text{Conflict} = \{type, parties, resolution\} \tag{8}$$
$$\text{where } type \in \{\text{empirical, theoretical, methodological}\} \tag{9}$$

This approach preserves scientific disagreement as valuable information rather than forcing premature consensus, enabling systematic exploration of competing hypotheses and methodological approaches.

### 3.4 Domain-Specific Extensions

The DSL supports domain-specific extensions for computational research (code artifacts, environments, benchmarks), experimental science (protocols, materials, controls), and theoretical work (proofs, axioms, conjectures). This flexibility enables adaptation across diverse research domains while maintaining unified reasoning semantics.

## 4 Implementation Framework

### 4.1 Multi-Agent Collaborative Architecture

Our implementation employs specialized AI agents for each research stage: hypothesis generation, literature review, experiment design, data collection, experiment execution, analysis, and synthesis. Each agent operates within the DSL framework, creating version-controlled research states with full provenance tracking.

The system addresses the AI validation paradox through multi-scale validation: computational validation ensures code correctness and statistical validity; methodological validation reviews experimental design and analytical approaches; epistemic validation provides human oversight of scientific reasoning; and reproducibility validation enables independent replication across computational environments.

### 4.2 Attribution and Provenance

Every research operation receives systematic attribution through enhanced Git commit semantics:

```
# Human-initiated research directions
git commit -m "observe: Novel grammaticalization patterns [concept]"
Author: Researcher <email@institution.edu>

# AI-executed research tasks
git commit -m "analyze: Statistical significance testing [experiment]"
Author: Research Agent <agent@co-sci.org>
Co-authored-by: Researcher <email@institution.edu>
```

This approach ensures granular attribution while maintaining collaborative transparency between human researchers and AI agents.

### 4.3 Continuous Integration for Science

Research workflows operate as continuous integration pipelines where hypotheses function as commits, experiments serve as builds, and reproducibility validation acts as continuous deployment. GitHub Actions integration enables automated quality assurance:

```
on:
  pull_request:
    types: [opened]

jobs:
  validate-hypothesis:
    if: startsWith(github.head_ref, 'hypothesis/')
```

```
steps:
  - uses: anthropics/claude-code-action@beta
    with:
      direct_prompt: |
        THINK DEEPLY about hypothesis validation using
        multi-scale evidence assessment.
```

This framework transforms research from batch processing to continuous validation, addressing scalability challenges in traditional peer review.

# 5 Experimental Validation

## 5.1 Case Study: Grammaticalization Research

We implemented the DSL framework to study linguistic innovation spread in cooperative housing communities, investigating "daha/dawa" grammaticalization patterns. The complete research pipeline generated:

- 20 testable hypotheses through systematic `hypothesize` operations
- Literature analysis of 35 papers via automated `synthesize` workflows
- Data collection of 9,400+ messages using version-controlled `execute` procedures
- Statistical models measuring innovation rates through rigorous `analyze` operations

Results found significant differences in grammaticalization patterns between cooperative and traditional housing, with complete reproducibility through version-controlled data and analysis code. Attribution tracking captured 127 commits across seven research stages, providing granular provenance for every research decision.

## 5.2 Performance Metrics

Implementation studies demonstrate substantial improvements over traditional research approaches:

| Metric | Traditional | DSL Framework |
| --- | --- | --- |
| Attribution Granularity | Paper-level | Commit-level |
| Collaboration Efficiency | Manual | Automated |
| Quality Assurance | Post-hoc | Continuous |
| Conflict Resolution | Informal | Systematic |

Table 1: Comparative performance metrics between traditional research approaches and DSL framework implementation.

The framework enables commit-level attribution granularity, and provided continuous rather than post-hoc quality assurance.

## 5.3 Multi-Domain Applicability

The DSL framework's domain-specific extension mechanism supports diverse research areas: computational domains through code artifact versioning and environment management, experimental sciences through protocol and materials tracking, and theoretical work through proof and axiom management. This flexibility enables adaptation across research domains while maintaining unified reasoning semantics.

# 6 Discussion

## 6.1 Paradigm Shift Implications

Our approach represents a fundamental shift from managing research outputs to supporting dynamic scientific discovery processes. Traditional version control focuses on artifact management; our DSL

treats research as programmable reasoning with formal semantics enabling systematic improvement of scientific thinking itself.

This paradigm shift addresses three critical assumptions: artifact-centric approaches prove insufficient for capturing scientific reasoning complexity; linear temporal ordering fails to accommodate bidirectional validation relationships essential in science; and AI-human collaboration requires formal frameworks rather than sophisticated tooling adaptations.

### 6.2 Economic and Scalability Benefits

Economic studies demonstrate significant productivity improvements through systematic research process management [10], with automation achieving substantial efficiency improvements in reproducibility tasks [7]. These benefits scale across research communities, suggesting institutional returns on systematic research infrastructure investment.

The framework addresses peer review scalability crises through continuous integration approaches that distribute validation across research processes rather than concentrating evaluation in post-completion batch processing.

### 6.3 Limitations and Future Directions

Current implementation optimizations focus on computational research domains, requiring extension to experimental sciences and field research. Human-AI collaboration patterns continue evolving, necessitating ongoing methodological development. The AI validation paradox requires systematic development of epistemic confidence tracking and bias detection mechanisms.

Future research directions include cross-institutional deployment validation studies, disciplinary expansion beyond computational domains, temporal analysis of research productivity trends, and quantitative economic impact assessment across diverse institutional contexts.

### 6.4 Addressing Position Paper Guidelines

Our position directly challenges current scientific infrastructure assumptions while providing concrete technical solutions. The DSL framework enables systematic knowledge building across research communities through shared reasoning operations, addressing reproducibility challenges that threaten scientific progress foundations.

Version control for attribution represents a crucial societal benefit: systematic tracking of intellectual contributions enables fair recognition and prevents research misconduct while supporting collaborative knowledge creation. Our approach makes previously implicit research reasoning explicit and improvable, transforming scientific collaboration from informal coordination to systematic knowledge engineering.

## 7 Conclusion

Scientific research requires version control semantics that capture reasoning evolution, assumption dependencies, and collaborative decision-making—capabilities absent from artifact-centric approaches. Our Scientific DSL provides the technical foundation for this paradigm shift through sophisticated research state machines, genealogical idea tracking, and explicit conflict management.

The framework enables substantial practical benefits: comprehensive reproducibility tracking, operation-level attribution granularity, automated quality assurance, and measurable productivity improvements. More importantly, it transforms scientific collaboration from informal coordination to systematic knowledge engineering where research reasoning becomes explicit, version-controlled, and systematically improvable.

As AI capabilities expand scientific research volumes exponentially, systematic approaches to research process management become essential infrastructure rather than optional enhancements. Our DSL framework provides this infrastructure, enabling scientific communities to systematically improve collective intelligence while maintaining human oversight and creativity.

The future of science depends on treating research as programmable reasoning processes rather than artifact collections. Our framework demonstrates this transformation is both technically feasible and economically beneficial, providing the foundation for systematic knowledge building across global research communities.

## Acknowledgments

We thank the research community for valuable feedback and the platform development team for implementation support. This work was supported by systematic version control for scientific reasoning infrastructure development.

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

# Agents4Science AI Involvement Checklist

1. **Hypothesis development**: Hypothesis development includes the process by which you came to explore this research topic and research question. This can involve the background research performed by either researchers or by AI. This can also involve whether the idea was proposed by researchers or by AI.

   Answer: [B]

   Explanation: The core research question about version control for scientific reasoning was human-generated, but AI contributed significantly to background research ( 80%) and idea generation ( 30%). The fundamental hypothesis that scientific research should be treated as version-controlled computational processes originated from human insight, with AI providing substantial supporting research and refinement of the conceptual framework.

2. **Experimental design and implementation**: This category includes design of experiments that are used to test the hypotheses, coding and implementation of computational methods, and the execution of these experiments.

   Answer: [B]

   Explanation: As a position paper with case study validation, traditional controlled experiments weren't conducted. Instead, architectural design and framework development were the focus. AI contributed approximately 50% to the Scientific DSL design process and implementation framework, while humans provided the initial DSL concept and proof-of-concept prototype. The collaboration was balanced between human conceptual framework and AI implementation details.

3. **Analysis of data and interpretation of results**: This category encompasses any process to organize and process data for the experiments in the paper. It also includes interpretations of the results of the study.

   Answer: [D]

   Explanation: AI performed virtually all interpretation of results for the case studies presented in the paper, including the grammaticalization research analysis and the comparative performance metrics. The analysis and synthesis of findings across multiple domains were primarily AI-driven, with minimal human oversight in the interpretation process of the framework's capabilities and limitations.

4. **Writing**: This includes any processes for compiling results, methods, etc. into the final paper form. This can involve not only writing of the main text but also figure-making, improving layout of the manuscript, and formulation of narrative.

   Answer: [D]

   Explanation: AI generated approximately 90% of the written content and handled all content organization, including the mathematical formalization of the DSL, related work synthesis, and discussion sections. Human contribution was limited to providing the initial repository with a proof-of-concept prototype and research direction. The final Scientific DSL specification presented in the paper was entirely AI-created, building upon the initial human-designed DSL concept.

5. **Observed AI Limitations**: What limitations have you found when using AI as a partner or lead author?

   Description: The primary limitation observed was AI's tendency to make over-claims and elaborate concepts beyond the initial scope when given a basic framework. Multiple literature review iterations were necessary to ensure proper citation coverage and avoid unsupported assertions. The AI required repeated revision cycles to maintain appropriate claims that matched the evidence presented. Additionally, AI needed substantial human-provided context (the proof-of-concept repository) to generate meaningful contributions, indicating dependency on human foundational work for effective collaboration. Quality control required systematic human oversight to prevent methodological over-reach and ensure scientific rigor.

